# Malnutrition Risk: Four Year Outcomes from the Health, Work and Retirement Study 2014 to 2018

**DOI:** 10.3390/nu14112205

**Published:** 2022-05-26

**Authors:** Carol Wham, Jade Curnow, Andy Towers

**Affiliations:** 1School of Sport, Exercise and Nutrition, College of Health, Auckland Campus, Massey University, Auckland 0632, New Zealand; jncurnow@gmail.com; 2School of Health Sciences, Massey University, Palmerston North 4474, New Zealand; a.j.towers@massey.ac.nz

**Keywords:** nutrition risk, nutrition screening, New Zealand, older adults, community

## Abstract

This study aimed to determine four-year outcomes of community-living older adults identified at ‘nutrition risk’ in the 2014 Health, Work and Retirement Study. Nutrition risk was assessed using the validated Seniors in the Community: Risk Evaluation for Eating and Nutrition, (SCREENII-AB) by postal survey. Other measures included demographic, social and health characteristics. Physical and mental functioning and overall health-related quality of life were assessed using the 12-item Short Form Health Survey (SF-12v2). Depression was assessed using the verified shortened 10 item Center for Epidemiologic Studies Depression Scale (CES-D-10). Social provisions were determined with the 24-item Social Provisions Scale. Alcohol intake was determined by using the Alcohol Use Disorders Identification Test (AUDIT-C). Among 471 adults aged 49–87 years, 33.9% were at nutrition risk (SCREEN II-AB score ≤ 38). The direct effects of nutrition risk showed that significant differences between at-risk and not-at-risk groups at baseline remained at follow up. Over time, physical health and alcohol use scores reduced. Mental health improved over time for not-at-risk and remained static for those at-risk. Time had non-significant interactions and small effects on all other indicators. Findings highlight the importance of nutrition screening in primary care as nutrition risk factors persist over time.

## 1. Introduction

Globally, there has been a significant change in the population pyramid and the World Health Organisation has predicted that the population of adults older than 60 years is set to double by the year 2050 [1]. New Zealand’s life expectancy at birth (82.1 years), is among the top life expectancies of the Organisation for Economic Co-operation and Development (OECD) countries [2]. The older adults’ population over 65 years is currently 15% of the total New Zealand population [3]. Both the absolute number and percentage of populations aged 65+ and 85+ are expected to continue to rise, as there are no signs of deceleration of life expectancy. 

Disability and disease disproportionately affect New Zealand’s older people, with 45% of New Zealanders over the age of 65 living with at least one disability [4]. The consequence of accumulated health conditions is that it reduces people’s capacity for self-care, leading to an increased requirement for support from the health care system [5]. 

Older people are known to be at disproportionate risk of malnutrition and have an increased risk of developing health problems because of inadequate food and nutrition intake [6]. Decline in nutritional status is a modifiable lifestyle factor directly affecting development of malnutrition and in most cases is amenable to intervention [7,8]. It is important to identify nutritional vulnerability among community-dwelling older adults, so that preventative or supportive strategies may be implemented [9]. Early intervention in the malnutrition continuum through early screening may reduce negative consequences in individual and healthcare spending. 

Seniors in the Community Risk Evaluation for Eating and Nutrition for cognitively intact community-dwelling older adults is a valid and reliable 14-item (SCREENII) and an abbreviated 8-item (SCREEN II-AB) screening tool that assesses upstream and early determinants that can lead to impaired food intake and eventual malnutrition [10]. As part of the 2014 Health, Work and Retirement postal survey, we have previously described the malnutrition risk prevalence and associated health and social risk factors amongst community-living Māori and non-Māori older adults in New Zealand [11]. Of 2914 community-living older adults (749 indigenous Māori) aged 49–87 years who completed SCREEN II-AB, half (50.2%) of Māori and 32.7% of non-Māori were at malnutrition risk [11].

Several other studies have used SCREENII to investigate the prevalence of malnutrition risk in New Zealand older adults and suggest between 31–54% of community-dwelling older adults are at some degree of moderate to high malnutrition risk [12,13,14]. Whilst these studies have investigated malnutrition risk status and associated risk factors, the longer-term outcomes of malnutrition risk are unknown. The aim of this study was to investigate the four-year outcomes of malnutrition risk in community-dwelling older adults living in New Zealand. 

## 2. Materials and Methods

### 2.1. Study Design 

This longitudinal cohort study used data from the 2014 and 2018 collections of the Health, Work and Retirement Study (HWR). The HWR study is a government-funded study established in 2006 to examine community-dwelling, healthy-ageing in older adults in New Zealand via a postal survey. The survey is administered on a biennial basis to track experiences of health and its determinants. The 2014 wave of data collection additionally evaluated nutrition status using the abbreviated SCREENII-AB. The results of the SCREENII-AB were then used to quantify participants at nutrition risk. In 2018, the postal survey including SCREENII-AB was sent to the participants again. This study aim was to compare the four-year social and health-related outcomes among community-living older adults identified at malnutrition risk vs. those not at risk in the 2014 HWR study.

### 2.2. Participants and Recruitment

As part of the HWR study, participant recruitment was undertaken using equal probability random sampling from the 2006 electoral roll. The study design and sampling procedures have been described previously [15]. The baseline for the current study was in 2014 when an enlargement of the participant pool occurred, using methods identical to the initial recruitment. Exclusion criteria were, inability to contact, institutionalisation, or deceased [16]. The 2014 recruitment included an expanded age range (49 to 87 years) to ensure the population sample remained representative of the 50+ New Zealand population [16]. Oversampling of indigenous Māori participants increased the likelihood of a more representative population sample and was based on the Māori-descent indicator on the electoral roll. This study excluded participants if they had not completed the SCREENII-AB nutrition assessment in the 2014 postal survey. The total number of participants in the current study was 2405 at baseline, with 1471 participants included in the 2018 four-year follow up. The study was approved by the Massey University Human Ethics Committee (MUHEC) as a low-risk research project.

### 2.3. Measures

#### 2.3.1. Demographic Characteristics

We identified participants age through self-reported birth date. Key sociodemographic characteristics were categorised based on response to survey questions, i.e., gender (male/female/gender diverse), marital status (partnered/unpartnered), education qualifications (none/secondary/post-secondary), household composition (living alone/living with others) and residential description (standalone or detached/joined to one or more other household/unit-or-villa in a retirement village/other including moveable dwelling, or rest home). 

#### 2.3.2. Physical and Mental Health

The SF-12 was used to provide a view of health-related quality of life of the participants based on their perceived experience, knowledge, and awareness of their personal, physical, mental, and emotional status [17]. The scale presents two summary scores: physical and mental health related quality of life. The maximum score is 100; any score lower than 40 is indicative of perception of poor health and above 60 is indicative of perception of reasonable and better health. The SF-12 has been validated and is considered a reliable questionnaire in community-dwelling older adult populations worldwide [17].

#### 2.3.3. Depression

Respondent’s depression was assessed using the verified shortened 10 item Center for Epidemiologic Studies Depression Scale (CES-D-10) [18]. This self-rated survey asks participants how often in the past week they have experienced symptoms of depression and loneliness. Scores range from 0–30, with any score above or equal to 10 indicating depression. This measure has been validated for community-dwelling older adults as sensitive and specific, and no discrimination in gender, race, ethnicity, or education level [19].

#### 2.3.4. Hazardous Alcohol Use

The WHO Alcohol Use Disorders Identification Test—Consumption sub-scale (AUDIT-C): [20] was used to assess participants’ hazardous drinking with an added item ‘Have you ever drunk alcohol in the past?’ to define lifetime non-drinkers from non-drinkers with a history of consumption. The classifications for alcohol consumption were lifetime abstainer, current non-drinker, light drinker (AUDIT-C score: 0–3), moderate drinker (AUDIT-C score: 4–7) and heavy drinker (AUDIT-C score: 8–12). Hazardous alcohol use was defined as an audit-C score of 8 or more. There is a lack of commonality in the scoring thresholds for the AUDIT C tool and it may overestimate hazardous drinking by 33% when used as a standard threshold for older adults [21]. This must be kept in mind when interpreting the results from this study. The tool is best used as a screening tool for further screening as it is brief, easy to administer, and produces generally sensitive results for screening for hazardous alcohol use in community-dwelling older adults [22].

#### 2.3.5. Social Connectedness

The 24-item Social Provisions Scale was used to assess the existence of meaningful social connectedness [23]. Although six different social “provisions” are needed for individuals to feel adequately supported and to avoid loneliness, only the “attachment” and “social integration” subscales were used in this study. The “attachment” subscale is referred to as “loneliness” throughout this study. Each subscale has four questionnaire items: two worded positively and two worded negatively. The scores for the four questionnaire items are added and the total score for each subscale ranges from 4 to 16. The social provisions scale has been used worldwide and is recognised as an easy to use, reliable, and valid tool to assess social connectedness [24]. 

#### 2.3.6. Nutrition Status

The 2014 HWR survey assessed participant nutrition status using the abbreviated Seniors in the Community Risk Evaluation for Eating and Nutrition questionnaire (SCREENII-AB) [10]. The SCREENII-AB is self-administered and specifically designed for use amongst community-dwelling older adults. The 8-item questionnaire assesses participant appetite, dysphagia, fruit and vegetable servings, fluid intake, meal sharing, meal preparation, meal skipping and weight change giving a score from 0–48. Scores below and equal to 38 indicate nutrition risk and scores above 38 deem an individual not at risk [10]. 

### 2.4. Statistical Analysis

SPSS statistical software (Version 27, SPSS Inc., Chicago, IL, USA) was used to analyse the data. Descriptive statistics described the following variables: age group, gender, relationship status, educational qualifications, household composition, residence type and malnutrition risk status (Table 1). Firstly, baseline differences were explored between the risk groups. Nutrition risk (SCREENII-AB) was categorised as not-at-risk (>38), and at-risk (≤38). Scale scores for SF-12 physical health, SF-12 mental health, CES-D, AUDIT C and items from Social Provisions were displayed as means for those at-risk and those not-at-risk. *p* values were used, and a Cohens d efficient was calculated to test for significant difference using a *t*-test (Table 2). Secondly, differences between 2014 data and 2018 data were explored for each scale score. Means and standard deviations for key outcome measures were calculated for at-risk and not-at-risk groups for 2014 and 2018 (Figure 1, Figure 2, Figure 3, Figure 4, Figure 5 and Figure 6). Finally, outcome measures were analysed using separate repeated-measures multivariate analyses of variance (RM-MANOVAs) to assess statistical difference. The RM-MANOVA investigated the main effect of Time (two levels: 2014 vs. 2018) and the interaction effect of being at nutrition risk (two levels: at risk vs. not at risk) by Time (or Nutrition Risk*Time) on the outcomes of being at nutrition risk, which was indicated by four outcomes (i.e., SF-12 physical health, SF-12 mental health, CES-D, AUDIT C, social provisions subscale scores) (Table 3).

## 3. Results

### 3.1. Participants

An overview of the participants’ sociodemographic characteristics categorised by malnutrition risk status for men and women in 2014 HWR is provided in Table 1. From the survey, 2405 completed SCREENII-AB. Of the participants, 44.3% were men, 76.2% were partnered, and most (87.2%) lived in a standalone or detached house. Many participants (80.9%) lived with others: either a partner, child(ren), grandchild(ren), flatmate or boarder. Overall, 33.9% of the participants were at nutrition risk (SCREEN II-AB score ≤ 38). The mean SCREEN II-AB score was 39.4, and women (36.8%) had a higher proportion of nutrition risk than men (30.2%). 

### 3.2. Exploring the Baseline Differences

Table 2 outlines the key outcome measure scores at baseline (2014). At baseline, those who were at malnutrition-risk had poorer scores for SF-12 physical and mental health than those not-at-risk (*p* ≤ 0.001; d = 0.61; *p* ≤ 0.001; d = 0.65). This group were also found to have higher depression scores (*p* ≤ 0.001, d = 0.61). Cohens d result indicates that the differences in physical and mental health, and depression between the risk groups, had a medium effect.

In 2014, neither groups’ alcohol use was scored as hazardous (at-risk 3.37, not-at-risk 3.57). There was no significant difference in hazardous alcohol use at baseline between the nutrition-risk and not-at-risk groups (*p* = 0.07). This null effect is supported by a very low Cohens d co-efficient. At baseline, the at-risk group had lower scores for social connections than the not-at-risk group (*p* ≥ 0.001). Specifically, both “attachment” and “social integration” scores were significantly higher for not-at-risk than for at-risk, with medium effect sizes. 

### 3.3. Exploring Differences between Baseline and Four-Year Follow-Up

We did not explore change over time in risk categorisation; instead, the intent was to understand the chronic outcomes of malnutrition categorisation in 2014. Figure 1, Figure 2, Figure 3, Figure 4, Figure 5 and Figure 6 illustrate change over time for each of the key outcome measures for the at-risk and not-at-risk groups (with 95% CI for 2014 and 2018).

Over the four-year follow-up period, the nutrition-risk group scores for SF-12 mental health scores increased (Figure 1), whereas SF-12 physical health scores showed a decline (Figure 2). Depression scores had increased for the not-at-risk group, whilst the at-risk group scores were similar to baseline (Figure 3).

Figure 4 shows hazardous drinking dropped marginally from baseline to follow up in the at-risk group, whilst scores dropped more considerably over the four years in the not-at-risk group. Both Attachment and Social Integration marginal means remained parallel between the at-risk and not-at-risk groups over the four years (Figure 5 and Figure 6).

### 3.4. Assessing Statistically Significant Change

Whilst Figure 1, Figure 2, Figure 3, Figure 4, Figure 5 and Figure 6 suggest a significant change in sub-scores over time and differential effects for nutrition risk groups, it was essential to assess if these differences were significantly different.

A RM-MANOVA was undertaken to assess the differences in key scores between baseline and outcome for these two groups. This analysis explored the effect of time (baseline to follow up), nutrition risk (at-risk and not-at-risk) and the potential interaction between time and nutrition risk. The results of this analysis are presented in Table 3. 

First, the direct effects of nutrition risk showed that significant differences between at-risk and not-at-risk groups at baseline remained at follow up. 

Second, the direct effect of time showed a significant change in the scores for SF-12 Physical health, SF-12 Mental Health and hazardous alcohol use. Over time, physical health scores and hazardous alcohol use scores reduced, and mental health scores increased. Further, effect size indicators suggested the difference in physical health and mental health between malnutrition risk groups was large. 

Third, there was a significant time*malnutrition risk interaction effect on SF-12 mental health and non-significant interactions and small effects on all other indicators (Table 3). Specifically, while mental health increased over time for not-at-risk, it remained static for the at-risk group.

## 4. Discussion

This study found a third of the participants were at risk for malnutrition at baseline and direct effects of malnutrition risk remained at follow up. In 2014, we observed those at risk had worse self-reported physical and mental health, higher rates of depression, lower social connectedness and higher rates of hazardous drinking compared to those not-at-risk. Lower self-reported physical health has previously been reported among older adults at risk of malnutrition in Australia [25], Singapore [26], Sweden [27] and Taiwan [28]. Similarly, lower physical health-related quality of life was found to be independently associated with risk of malnutrition among community-dwelling older adults in New Zealand [14]. Our finding that the at-risk group had worse self-reported mental health is consistent with similar observations in Australia, Sweden and Nepal [25,27,29]. The 2014 baseline findings also showed that results for both social connections’ subscales were worse for the at-risk group than the not-at-risk group. Studies exploring this relationship, using the same tool as the current study, lacking links between social isolation and malnutrition risk in older adults are well established [30,31,32]. Having the opportunity to share dining experiences and interact at mealtimes increases food intake, which is positively correlated with nutritional quality as well as health outcomes [33,34]. At baseline, we found depression was also more prevalent in those at malnutrition risk than those not-at-risk, which supports findings from studies in Japan [35], Singapore [26], Taiwan [28] and New Zealand [14]. Although these studies used the Geriatric Depression Scale, this has the same specificity and sensitivity to the CES-D used in the current study [36]. While we observed no significant difference in drinking between nutrition risk groups, evidence suggests older adults who are at malnutrition risk are more likely to have light alcohol use (drink less than four times a week) than those not at risk [14,31] but this was not supported by our findings. *Changes at four years follow up*.

The current study showed that changes across time in health outcomes only occurred for physical health, mental health and hazardous alcohol use. We found the baseline difference of self-reported physical health between the risk groups persisted over time and the difference between at-risk and not at-risk increased. This is a unique finding, given the paucity of research exploring this relationship over time. One prospective study in Canada found that among those at low malnutrition risk, only poor self-rated physical health was a predictor of elevated risk at one year follow up [37]. 

We found mental health remained the same as it was at baseline among those not-at-risk and improved among the at-risk group at follow up. This may be the effect of interaction with supporting health services, but this is only speculative. In Canadian older adults, mental health remained the same at one year follow up for those at low malnutrition risk with no improvement in the at-risk group observed [37]. Social connectedness (i.e., measures of loneliness and attachment) were both worse for the at-risk group over time than the not at-risk group in this study. Further, the gap between the two nutrition risk groups did not change over time. This finding is supported by the Canadian prospective study with a similar population, where the level of satisfaction in social support over a one year follow up remained the same for both groups [37]. 

Findings indicate the at-risk group were significantly more depressed than the not-at-risk group at baseline and this persisted over the four-year period. We speculate that time is not a factor that influences depression in older adults, as there are many factors that influence the likelihood of depressive symptoms and may be a cause of weight loss and malnutrition [38]. There is currently a paucity of literature on the relationship between malnutrition and hazardous drinking. A prospective study of 579 home-living older people also found that no alcohol use was associated with malnutrition, whereas light alcohol use was not [39]. This indicates that light alcohol use may be protective from nutrition risk in community-dwelling older adults. 

Overall, the study findings illustrate that over time, the baseline differences between the risk-groups for most health outcomes remain the same. For both groups, self-reported physical health significantly decreased over time. Mental health, however, increased over time in the nutrition risk group, and hazardous drinking decreased for the at-risk group and remained the same for the not-at-risk group. 

A strength of this study is the investigation of health and social outcomes associated with malnutrition risk over a four-year period, making this study the first of its kind. A major limitation relates to the longitudinal observational design of the study where nutritional intervention could not be provided for participants identified at risk of malnutrition in the 2014 postal survey. Further limitations include the study population of older people, which does not extend beyond age of 87 years, thus missing wider representation of adults in advanced age. Although the HWR study sample was designed to be representative, wave on wave variation in response rates may diminish representativeness. With the self-completed questionnaire, there may be inaccuracies in self-reports, as these rely on memory and accurate recollection. Those with poor literacy skills, visual impairments or functional limitations may have required a scribe, increasing response bias. The use of the 12-item Short-Form Health Survey may have limitations for indigenous Māori for whom health has a broader perspective [40].

## 5. Conclusions

To conclude, this study’s baseline analysis found older adults at malnutrition risk have worse self-reported physical and mental health, higher rates of depression, lower social connectedness, and higher rates of hazardous drinking compared to those not-at-risk. Follow up over time suggested that most of the distinctions between at-risk and not at-risk groups remained the same and were not resolved with the passage of time. The only caveat being mental health. 

Collectively, these findings can assist in designing nutrition-related interventions aimed at supporting ageing in place. Depression has an adverse impact on appetite, food intake and physical capacity [41] and lower social connectiveness suggests older adults at malnutrition risk may benefit from the social facilitation of eating in the presence of others. Mealtime companionship is a better predictor of caloric intake than marital status [34]. In particular, social supports that provide a sense of close social connection as opposed to simple emotional attachments to relatives, friends, and community, are significantly associated with higher caloric intakes [42]. Screening and/or assessment of nutrition and physical performance status are cost effective and efficient procedures, which may enable timely dietary interventions and for encouraging older adults to share meals with others.

This is the first study to analyse the direct impact of time and nutrition risk as a combined factor, which had a direct influence on increasing mental health. Findings highlight the utility of nutrition screening in primary care so that older adults at nutrition-risk are identified for early intervention. 

## Figures and Tables

**Figure 1 nutrients-14-02205-f001:**
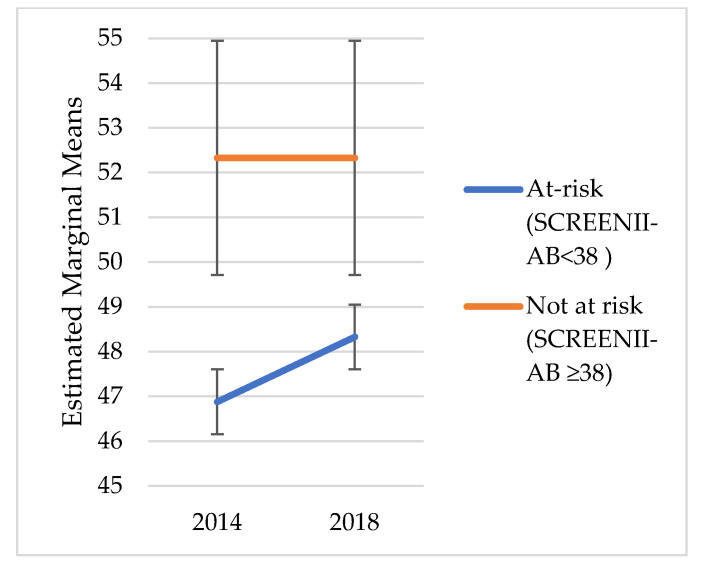
SF-12 Mental Health Estimated Marginal Means.

**Figure 2 nutrients-14-02205-f002:**
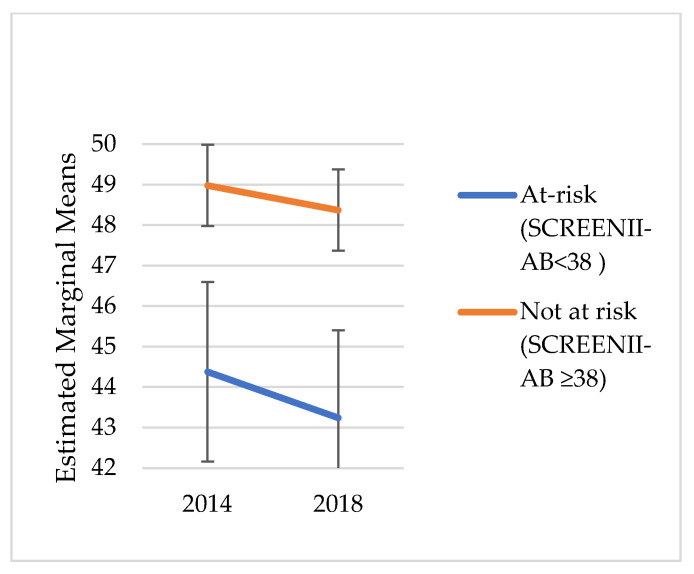
SF-12 Physical Health Estimated Marginal Means.

**Figure 3 nutrients-14-02205-f003:**
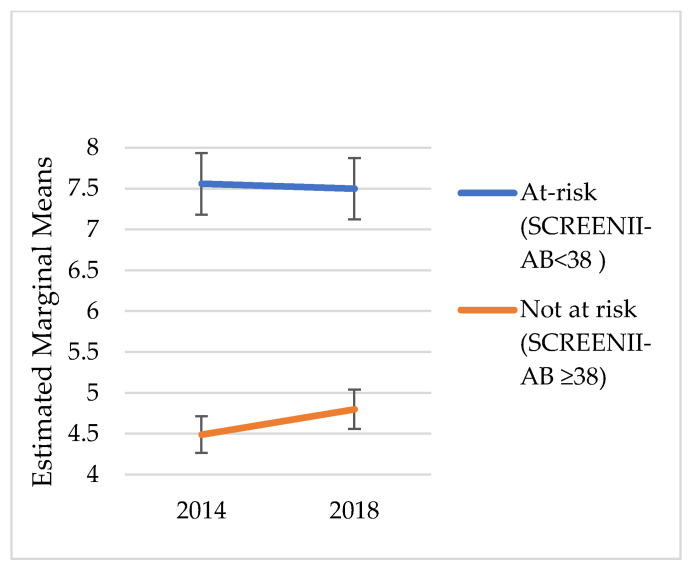
Depression Estimated Marginal Means.

**Figure 4 nutrients-14-02205-f004:**
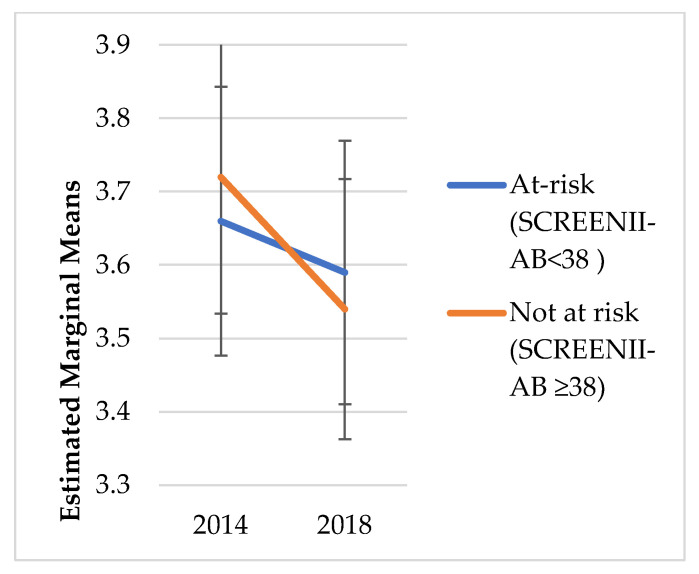
Hazardous Drinking Estimated Marginal Means.

**Figure 5 nutrients-14-02205-f005:**
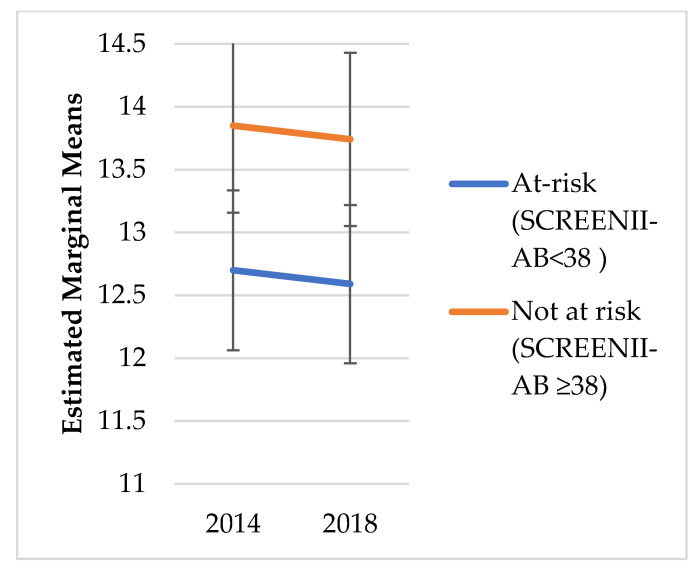
Attachment Estimated Marginal Means.

**Figure 6 nutrients-14-02205-f006:**
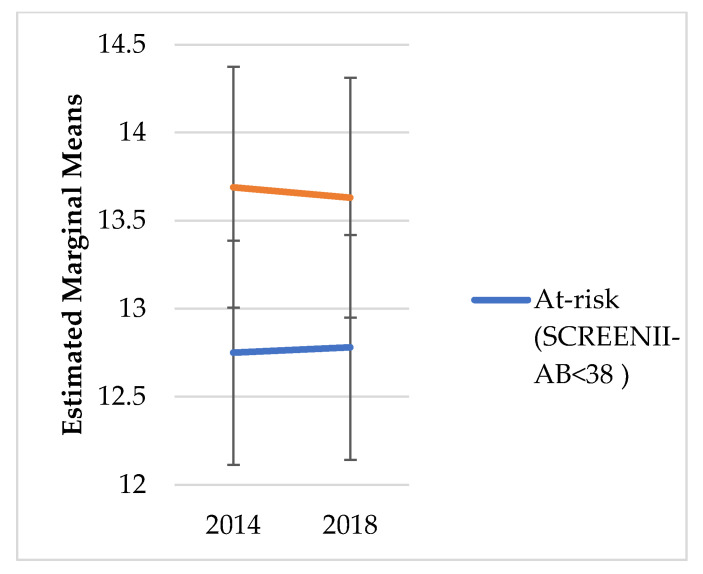
Social Integration Estimated Marginal Means.

**Table 1 nutrients-14-02205-t001:** Participants’ sociodemographic characteristics categorised by malnutrition risk in 2014.

	Total *n* = 2405	At-Risk *n* (%) SCREENII-AB < 38 782 (33.9)	Not at-Risk *n* (%) SCREENII-AB ≥ 38 1528 (66.1)
Age (49–87)			
Mean (SD)	66 (6.2)	65.8 (6.3)	66.2 (6.1)
Sex			
Male	1066 (44.3)	308 (39.4)	713 (46.7)
Female	1339 (55.7)	474 (60.4)	815 (53.3)
Ethnicity			
NZ European	1598 (71)	459 (60.2)	1139 (76.6)
Māori	537 (23.9)	252 (33)	285 (19.1)
Other	115 (5.1)	52 (6.8)	63 (4.2)
Marital status			
Partnered	1805 (76.2)	486 (63.4)	1249 (82.9)
Un-partnered	564 (23.8)	281 (36.6)	258 (17.1)
Education			
No qualifications	500 (27.1)	207 (27)	257 (17)
Secondary school	528 (28.6)	169 (22.1)	345 (22.9)
Post-secondary	819 (44.3)	390 (50.9)	907 (60.1)
Household composition			
Living alone	546 (19.1)	200 (26.2)	218 (14.4)
Living with others	2315 (80.9)	563 (73.8)	1291 (85.6)
Residence type			
Standalone or detached	2045 (87.2)	641 (86.3)	1332 (90.5)
Joined household	207 (8.8)	82 (11)	112 (7.6)
Retirement village	38 (1.6)	12 (1.6)	23 (1.9)
Other, including moveable dwelling, or rest home	56 (2.3)	8 (0.54)	5 (0.3)
Hospital usage in last 12 months			
Yes	811 (36.1)	315 (41.4)	496 (33.3)
No	1438 (63.9)	445 (58.6)	993 (66.7)

SCREEN II-AB, Seniors in the Community: Risk Evaluation for Eating and Nutrition, Version II, abbreviated values are count (percent); Missing data: Percentages may not always add up to 100% for each column as there is missing data for approximately 4.6% of subjects across various questions/sub sections within the 2014 HWR questionnaire.

**Table 2 nutrients-14-02205-t002:** Key outcome measure scores of at-risk and not at-risk groups in 2014.

	Scale Scores		
	At-Risk *n (%)* SCREENII-AB < 38 782 (33.9)	Not at-Risk *n (%)* SCREENII-AB ≥ 38 1528 (66.1)	Test for Significant Difference
	Mean (SD)	Mean (SD)	
SF-12 Physical health	42.94 (11.06)	48.56 (8.27)	*p* ≤ 0.001; d = 0.61
SF-12 Mental health	46.25 (11.29)	52.29 (8.05)	*p* ≤ 0.001; d = 0.65
Depression	8.05 (5.15)	4.65 (3.80)	*p* ≤ 0.001; d = 0.79
Hazardous alcohol use	3.37 (2.42)	3.57 (2.06)	*p* = 0.07; d = 0.09
Social connections			
Attachment	12.67 (2.35)	13.76 (2.09)	*p* ≤ 0.001; d = 0.50
Social integration	12.7 (1.92)	13.6 (1.79)	*p* ≤ 0.001; d = 0.49

SCREEN II-AB, Seniors in the Community: Risk Evaluation for Eating and Nutrition, Version II, abbreviated, SF-12, 12-item Short Form Health Survey. Significant differences between the two nutrition risk groups as determined by independent samples *t*-test (2-tailed). Cohens d values around 0.2 indicate a small effect size, values around 0.5 indicate a medium effect size, and values around 0.8 indicate a large effect size. Codes: *p* ≤ 0.05 = significant difference d = effect size.

**Table 3 nutrients-14-02205-t003:** Univariate main and interaction effects: malnutrition risk.

Effect	Univariate Outcome	*N*	*df*	*F*	*p* *	η^2^*p*
Time	SF-12 Physical health	1471	1, 1469	13.65	<0.001 *	0.009
	SF-12 Mental health	1471	1, 1469	8.14	0.004 *	0.006
	Depression	1471	1, 1469	1.195	0.275	0.001
	Hazardous alcohol use	1471	1, 1469	12.08	0.001 *	0.008
	Attachment	1471	1, 1469	4.23	0.40	0.003
	Social integration	1471	1, 1469	0.14	0.712	<0.001
Malnutrition risk	SF-12 Physical health	1471	1, 1469	106.03	<0.001 *	0.67
	SF-12 Mental health	1471	1, 1469	112.43	<0.001 *	0.71
	Depression	1471	1, 1469	174.36	<0.001 *	0.106
	Hazardous alcohol use	1471	1, 1469	0.004	0.952	<0.001
	Attachment	1471	1, 1469	113.52	<0.001 *	0.072
	Social integration	1471	1, 1469	93.52	<0.001 *	0.06
Time*Malnutrition risk	SF-12 Physical health	1471	1, 1469	1.26	0.263	0.001
	SF-12 Mental health	1471	1, 1469	8.19	0.004 *	0.006
	Depression	1471	1, 1469	2.65	0.104	0.002
	Hazardous alcohol use	1471	1, 1469	2.66	0.103	0.002
	Attachment	1471	1, 1469	0.008	0.929	<0.001
	Social integration	1471	1, 1469	0.79	0.374	0.001

* *p* ≤ 0.05 = significant difference.

## Data Availability

The data presented in this study are available on request from the corresponding author.

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
