# Peer review of "Malnutrition Risk: Four Year Outcomes from the Health, Work and Retirement Study 2014 to 2018"

_nutrients, 2022, doi:10.3390/nu14112205_

Round 1
Reviewer 1 Report
Review of “Malnutrition Risk: Four Year Outcomes from the Health, Work 2 and Retirement Study 2014 to 2018.”
This is an interesting longitudinal study looking at aassociations of the “Seniors in the Community: Risk Evaluation for Eating and Nutrition, abbrevi- 11 ated version (SCREENII-AB)” and health outcomes. Overall it provides important new insights, but I do have some comments:
- The authors should report quantitative results in the abstract
- The statistical approach is quite complex and may be hard to understand for the reader. In addition it remans unclear what the primary outcome measure was and the main analysis? It would be good to state the main hypothesis, the main outcome and the main predictor – and also to present this analysis in the abstract
- In addition to the repeated-measures multivariate analyses of variance, it would be good to present a more simple regression model looking at baseline nutritional risk and clinical outcome over time – adjusted for important baseline factors. Do the authors have data on mortality ? Also low QoL defined by the SF12 could be used here, or similar
- One important variable seems to be counselling of participants and start of nutritional interventions. Do the authors have data on these kex variables? If not – this should be discussed as a major limitation.
- The main conclusion of the report is that “Without treatment, most aspects of health will not improve…”. Please provide some references for this statement – do we have enough evidence that malnutrition can be reversed by nutritional interventions (given your findings that most patients with malnutrition have alcohol abuse and mental disorders)?
Author Response
Please provide a point-by-point response to the reviewer’s comments
Reviewer 1
This is an interesting longitudinal study looking at aassociations of the “Seniors in the Community: Risk Evaluation for Eating and Nutrition, abbrevi- 11 ated version (SCREENII-AB)” and health outcomes. Overall it provides important new insights, but I do have some comments:
Thank you, your insights are appreciated.
- The authors should report quantitative results in the abstract
Response: Thank you for your comment. We have discussed your suggestion. This is very challenging when the abstract should be 200 words maximum. The guidelines suggest the abstract should be an objective representation of the article. Guidelines further state the results should summarize the article's main findings.
The statistical approach is quite complex and may be hard to understand for the reader. In addition it remans unclear what the primary outcome measure was and the main analysis? It would be good to state the main hypothesis, the main outcome and the main predictor – and also to present this analysis in the abstract
Response: Again, the abstract maximum is 200 words. We will make it clear the outcome measure was SCREENII-AB -where scores indicate whether an individual is at risk or not at nutrition risk. The main outcome was in the nutrition risk group or not. We were interested in what factors resulted in being at nutrition risk or not.
- In addition to the repeated-measures multivariate analyses of variance, it would be good to present a more simple regression model looking at baseline nutritional risk and clinical outcome over time – adjusted for important baseline factors. Do the authors have data on mortality ? Also low QoL defined by the SF12 could be used here, or similar
Response: The repeated measures MANOVA explores the role of baseline factors while controlling for change over time. No mortality data was available. SF12 was included as a predictive factor, the outcome was malnourished not quality of life.
- One important variable seems to be counselling of participants and start of nutritional interventions. Do the authors have data on these kex variables? If not – this should be discussed as a major limitation.
Response: A further reference (10) has been included in the methods to explain this is a longitudinal observational study that aims to characterize health and well-being of older adults in New Zealand and to identify key determinants. This is a longitudinal postal survey which began in 2006, and follow-up surveys are administered on a biennial basis. The New Zealand national electoral roll is used as the sampling frame for cohort recruitment. Previous participants are considered to remain “active” in the study and resurveyed if they (1) have a New Zealand Postal address; (2) are not known to be deceased; (3) are not otherwise known to be lost to contact and (4) have not contacted the study to withdraw their participation. New participant samples are randomly selected from adults on the electoral roll who meet the inclusion criteria. This means participants identified at risk of malnutrition in 2014 were not contacted to receive nutrition assessment or intervention. This is now noted as a major limitation of the longitudinal study design as you suggest. See Line 236.
- The main conclusion of the report is that “Without treatment, most aspects of health will not improve…”. Please provide some references for this statement – do we have enough evidence that malnutrition can be reversed by nutritional interventions (given your findings that most patients with malnutrition have alcohol abuse and mental disorders)?
Response: Thank you for pointing this out. We agree that specific evidence is lacking, and this sentence is now deleted. The last sentence now reads “Findings highlight the utility of nutrition screening of older adults in primary care so that older adults at nutrition-risk are identified for early dietary intervention”.
Reviewer 2 Report
Abstract:
Remove word “abbreviated version” from line 11-12
Introduction:
Add reference in first paragraph of introduction
Since when NZ is facing issues related to population aging? I advised you try to write specifically, write year.
I have thoroughly read your introduction, and I feel that you need to rework. I have a few suggestions:
- In the first paragraph of your introduction, try to introduce about the topic, where you can present the global health and nutrition status among the elderly population. In the following paragraph present situation related to developed and developing countries. The focus of intervention for most of the developing countries is MNCH, while in developed nations, including NZ managing health finances and health care need of elderly population is a great concern.
- After presenting this global situation, try to present a paragraph enriched with epidemiological statistics and causes and present situation related to NZ and present the background of your previous work.
- After that write a rational together with study objectives
Methodology:
I read the methodology section, I counted from 2014 to 2018, it is 5 years NOT 4 year.
Section 2.1: In line 56, you already defined the abbreviation of Health, Work and Retirement (HWR), and in line 66 you again mention full form of HWR. I suggest you present abbreviation everywhere instead of full form. Because you already describe the abbreviation in line 56.
Section 2.2: In the exclusion criteria, one criterion is institutionalisation. I did not understand what do you mean by this? In line # 71, you mentioned age in the brackets, but you did not describe the unit, either it is year or months or days or weeks. You have described number of participants at baseline and at endline, and this is very important information for assessing the strength of your study. I advised you present what is loss to follow-up rate. Additionally, I also suggest you make a flow diagram, and present how many participants you had at baseline, when loss to follow-up occur and what were the reason at each point. I do understand, this would be bit time consuming for you, but if someone wants to replicate such type of study, then he/she should be aware about the hurdles you face while you were doing this study.
Section 2.3: What are the eight subclasses of SF-12 or SF-36? What is the reliability of SF-12? Try to explain further about SF-12, the questionnaire consists of 12 items. How did you score each item? Remove the example from line 129-133. You have written each sub-scale has four items, then after example you wrote each statement. Make it consistent. You can choose either statement or item in the following sentence.
In social connectedness, you wrote you measure attachment and social integration. Later, you described you analysed attachment, and refeered it to loneliness throughout the study. Then you described about the sub-scale. I did not understand was that for attachment or for both? I think you need to write this paragraph clearly.
I suggest you follow STROBES guidelines for writing the methodology and results of your study and keep all headings, which are in STROBES of longitudinal study. This will bring more clarity in your writing.
Section 2.4: I want to know how did you deal with missing variables? Because you had a loss to follow up at endline.
Results:
Table-2: Only present p-value and cohen-d value and remove t-value
Table 3- Add asterik sign (*) over the top of significant covariates and significant interactions. And also add a legend and write where asterik show significance
Section 3.3: Remove the line 226-229. “The difference between . . .. in 2014”. This is not result, this could be part of method or/and of discussion.
Line # 233-234 seem incomplete, I think you are referring to both group at risk and not at risk so complete the sentence.
I also suggest you to change the way of presentation, because I did not understand what
Figure-1 to Figure-6 are explaining. Either the difference is significant or not. Either it is improving or getting worse among two-groups. You can present it tabular form or think to present in any alternative fashion.
Section 3.4: All the content which you wrote here is more related to statistical method, rather than results. I suggest you present results in three paragraph, and describe the important findings in relation with time, then malnutrition, and then interaction.
Discussion:
I feel that discussion is fine. But you need to add some implications of this study in a separate paragraph? Moreover, how can we overcome the issues related to dementia among elderly population?
Author Response
Reviewer 2
Abstract:
Remove word “abbreviated version” from line 11-12
Response: thank you this is now removed
Introduction:
Add reference in first paragraph of introduction
Response: Thank you we have replaced the original reference with a more pertinent reference.
Since when NZ is facing issues related to population aging? I advised you try to write specifically, write year.
I have thoroughly read your introduction, and I feel that you need to rework. I have a few suggestions:
- In the first paragraph of your introduction, try to introduce about the topic, where you can present the global health and nutrition status among the elderly population. In the following paragraph present situation related to developed and developing countries. The focus of intervention for most of the developing countries is MNCH, while in developed nations, including NZ managing health finances and health care need of elderly population is a great concern.
After presenting this global situation, try to present a paragraph enriched with epidemiological statistics and causes and present situation related to NZ and present the background of your previous work.
- After that write a rational together with study objectives
Response: Thank you we have reworked this section following your suggestions.
Methodology:
I read the methodology section, I counted from 2014 to 2018, it is 5 years NOT 4 year.
Response: Thank you. This longitudinal survey has been administered on a biennial basis since 2006. Hence the survey is taken every 2 years and from the 2014 survey we have 4 years of data.
Section 2.1: In line 56, you already defined the abbreviation of Health, Work and Retirement (HWR), and in line 66 you again mention full form of HWR. I suggest you present abbreviation everywhere instead of full form. Because you already describe the abbreviation in line 56.
Response: Thank you abbreviated form is now provided.
Section 2.2: In the exclusion criteria, one criterion is institutionalisation. I did not understand what do you mean by this? In line # 71, you mentioned age in the brackets, but you did not describe the unit, either it is year or months or days or weeks. You have described number of participants at baseline and at endline, and this is very important information for assessing the strength of your study. I advised you present what is loss to follow-up rate. Additionally, I also suggest you make a flow diagram, and present how many participants you had at baseline, when loss to follow-up occur and what were the reason at each point. I do understand, this would be bit time consuming for you, but if someone wants to replicate such type of study, then he/she should be aware about the hurdles you face while you were doing this study.
Response: The HWR study data is collected by a voluntary postal survey. This is a standard longitudinal study. We have used a two-time comparison only where participants were only included if they completed the 2014 and 2018 waves. As stated above participants are resurveyed if they (1) have a New Zealand Postal address; (2) are not known to be deceased; (3) are not otherwise known to be lost to contact and (4) have not contacted the study to withdraw their participation. Hence if participants were institutionalised or required aged residential care between 2014 to 2018 they were excluded. The design of the study does not allow us to undertake an analysis of who was lost to the study and why. This applies to all HWR study publications.
Section 2.3: What are the eight subclasses of SF-12 or SF-36? What is the reliability of SF-12? Try to explain further about SF-12, the questionnaire consists of 12 items. How did you score each item? Remove the example from line 129-133. You have written each sub-scale has four items, then after example you wrote each statement. Make it consistent. You can choose either statement or item in the following sentence.
Response: Thank you the description of SF- 12 has been rewritten to address your concerns.
In social connectedness , you wrote you measure attachment and social integration. Later, you described you analysed attachment, and refeered it to loneliness throughout the study. Then you described about the sub-scale. I did not understand was that for attachment or for both? I think you need to write this paragraph clearly.
Response: Thank you the description for social connectedness has also been rewritten.
I suggest you follow STROBES guidelines for writing the methodology and results of your study and keep all headings, which are in STROBES of longitudinal study. This will bring more clarity in your writing.
Response: The STROBE checklist and guidelines have been followed.
Section 2.4: I want to know how did you deal with missing variables? Because you had a loss to follow up at endline.
Response: As above the study included 1471 participants in the 2018 four year follow up and there were missing data for approximately 4.6% of subjects across various questions/sub sections within the 2014 HWR questionnaire (see Table 1).
Results:
Table-2: Only present p-value and cohen-d value and remove t-value
Response: t-values are moved.
Table 3- Add asterik sign (*) over the top of significant covariates and significant interactions. And also add a legend and write where asterik show significance
Response: Added asterik sign (*) over the top of significant covariates
Section 3.3: Remove the line 226-229. “The difference between . . .. in 2014”. This is not result, this could be part of method or/and of discussion.
Response: As suggested have removed the lines 226-229
Line # 233-234 seem incomplete, I think you are referring to both group at risk and not at risk so complete the sentence.
Response: Yes have tried to clarify -referring to both group at risk and not at risk
I also suggest you to change the way of presentation, because I did not understand what
Figure-1 to Figure-6 are explaining. Either the difference is significant or not. Either it is improving or getting worse among two-groups. You can present it tabular form or think to present in any alternative fashion.
Response: Thank you. The wording was inaccurate and has been changed to reflect what the figures show. Figures 1-6 illustrate marginal means for the outcome measures for the at risk (blue) and not at-risk (yellow) groups with 95% CI for each year (2014 and 2018) in black lines. It is standard to illustrate graphically change over time.
Section 3.4: All the content which you wrote here is more related to statistical method, rather than results. I suggest you present results in three paragraph, and describe the important findings in relation with time, then malnutrition, and then interaction.
Response: This is now split into 3 paragraphs
Discussion:
I feel that discussion is fine. But you need to add some implications of this study in a separate paragraph? Moreover, how can we overcome the issues related to dementia among elderly population?
Response: Thank you this is a good suggestion. A further paragraph is now added.
Round 2
Reviewer 1 Report
thank you for revising the manuscript